# Viability of 4 Probiotic Bacteria Microencapsulated with Arrowroot Starch in the Simulated Gastrointestinal Tract (GIT) and Yoghurt

**DOI:** 10.3390/foods8050175

**Published:** 2019-05-24

**Authors:** Lesly Samedi, Albert Linton Charles

**Affiliations:** Department of Tropical Agriculture and International Cooperation, National Pingtung University of Science and Technology, 1 Shuefu Road, Neipu, Pingtung 0912, Taiwan; leslysamedi3@gmail.com

**Keywords:** microencapsulation, lyophilization, capsules, freeze-dried bacteria, arrowroot

## Abstract

Probiotic bacteria are usually encapsulated to increase their survival through passage of the simulated gastrointestinal tract (GIT). Four *Lactobacilli* were freeze-dried and encapsulated with maltodextrin (maltodextrin 1.25 g, whey 0.25 g, bacteria 0.5 g, and water 2 mL) and arrowroot starch (arrowroot 1.25 g, whey 0.25 g, bacteria 0.5 g, and water 2 mL). The effects of different coatings were evaluated for their viability in the GIT and yogurt. The findings indicated no significant differences at *p* > 0.05 in the survival of the encapsulated cells with increased concentrations of arrowroot and maltodextrin. The viability of the encapsulated bacteria was increased in the simulated GIT with high counts of 10^9^ cfu/mL after 30 min stiffening in 1 µm size beads. However, the bead fermented yogurt exhibited insignificant difference on the survivability of the organisms in a simulated GIT after 15 days. *Lactobacillus*
*plantarum*, *Weissela*
*paramesenteroides*, *Enterococcus*
*faecalis*, and *Lactobacillus*
*paraplantarum* showed a significant increase of viable cells at *p* > 0.05 after freeze-drying in comparison with free cells at high bile salt concentrations and low acidity. This study confirmed that arrowroot starch and maltodextrin combinations in encapsulation might be an effective method that could allow viable probiotic bacteria to reach the large intestine.

## 1. Introduction

Probiotics are live microbial preparations that are used as food additives and manifest health benefits in humans that include improving digestion and intestinal hygiene [1]. Cheese and yogurt are recognized as good vectors of probiotics in particular because of their wide consumption. Some researchers have suggested that the daily intake of probiotics must be between 10^8^ and 10^9^ cfu/day to have positive or benefit effect [2]. However, studies conducted on several commercial dairy products show low survivability of bacterial probiotics at ingestion. To develop probiotic dairy products with proven effects in humans, guaranteeing total consumer satisfaction in terms of health benefits, safety, quality, and practicality of use, it is therefore essential to maintain or even reinforce the viability and functionality of strains during all stages of the manufacturing and storage of products.

The popularity of the functional foods market is growing rapidly around the world, as consumers are increasing their expectations on foods that combine taste and health benefits. To meet this demand, bioactives have been developed in the past, but only a few that are useful in the encapsulation of probiotic ingredients have been added to food or nutraceutical products. In particular, probiotic bacteria have received considerable interest and their incorporation into food is growing [3]. The challenge of these functional foods is to preserve the functionality of the bacteria present and to ensure that they reach the site of their activity in sufficient quantity. Microencapsulation is a technique used to ensure the intestinal release of microorganisms, prevent their deterioration, improve their viability and, ultimately, reduce their interrelation with food constituents during ingestion. There are many encapsulation techniques for bacteria [4]. Likewise, many encapsulation matrices have been used to protect bioactive compounds, but only a small number are compatible with the viability of the organisms. The strategic choice of encapsulation methods and its matrix must make it possible to reach the desired characteristics for the micro-particles produced.

To increase their survival rate, probiotics must be ingested during mealtime, or coated in beads or by microcapsules [5]. The choice of vectors by which the probiotics are ingested is also important (tablets, gelatin capsules, fermented milk). Standard techniques for measuring the resistance of probiotic strains to gastrointestinal stress consist in evaluating their viability after stress in an agar count [6,7]. This technique has three major defects: it is long (24 to 72 h depending on the species), it underestimates the number of cells (chain problems in *Lactococci* for example); and it does not take into account viable cells that are not cultivable (which could return to an active physiological state under conditions other than culture conditions) [8]. So, while many authors are interested in the effects of stress on probiotics, they are most often attached to the study of cultivability and seldom to the maintenance of the properties of the microorganisms and generally are only concerned with one or two strains. Even if it is possible to distinguish more sensitive microbial groups than others, resistance to stress, strongly depends on strains [9].

Sultana reported that the incorporation of Hi-Maize^®^ starch improved encapsulation of *Lactobacillus acidophilus* and *Bifidobacterium* spp. in yoghurt, as compared to when the bacteria were encapsulated without the starch [10]. Habitually, 1–2% insoluble starch grains added to the probiotic–hydrocolloid precursor directly before the encapsulation process aim to further maintain the viability of probiotics [10,11]. Moreover, arrowroot starch, a commercially underexploited tuber starch but having potential digestive and medicinal properties, is a natural healer. It has a percentage digestibility of 30.07% after 30 min of incubation with the enzyme and 25.27% to 30.56% after extrusion. Its absorption index is 6.52 to 8.85 g gel/g dry sample, water solubility index 15.92% to 41.31%, and oil absorption index 0.50 to 1.70 g/g were higher for the extrudates compared to native starch of 1.81 g gel/g dry sample, 1.16% and 0.60 g/g, respectively [12]. Therefore, this research aimed to determine the viability of encapsulated bacteria in a simulated gastrointestinal tract (GIT) and in yogurt for 90 days of storage. Thus, the efficiency of encapsulation with prebiotics and freeze-drying methods on the survivability of *W. paramesenteroides* (CP023501.1), *L. paraplantarum* (AB362736.1), *E. faecalis* (HQ802261.1), and *L. plantarum* (MF369875.1) was examined in simulated GIT and yogurt after process and storage.

## 2. Materials and Methods

### 2.1. Bacteria, Growth Conditions, and Preparation of Cell Suspensions

Four strains of probiotics including *W. paramesenteroides* (CP023501.1), *L. paraplantarum* (AB362736.1), *E. faecalis* (HQ802261.1) and *L. plantarum* (MF369875.1) were selected for the study. The isolates were individually inoculated in 10 mL of sterile MRS. Furthermore, this sterile broth with the cell cultures was alimented anaerobically with sterile filter 0.05% *w*/*v* L-cysteine/hydrochloride (Sigma Chemical Co., Castle Hill, Sydney, Australia). The activation of the cultures was carried out in a triplicate run at 37 °C for 24 h [13]. Thus, the culture of the cells was performed in 500 mL of MRS broth for lyophilization; and these were collected using the Sorvall RT7 refrigerated centrifuge at 1600× *g* at 25 °C during 25 min. The cells were collected, washed, and then suspended in 90 mL Schott bottles of reconstituted sterile skimmed milk (14% *w*/*v*) [14].

### 2.2. Survival of Free and Freeze-Dried Strains in Low pH and Bile Salt Conditions

The encapsulated and free bacteria were added to non-fat milk, glucose, yeast extract and cysteine medium (12% non-fat skim milk, 2.0% glucose, 1.0% yeast extract, and 0.05% cysteine) (NGYC) that had been adjusted to pH 2.0, pH 3.0, or 6.5 (control) with 5 M HCl or 1 M NaOH in 10 mL aliquots. The samples were incubated at 37 °C for 3 h. An aliquot from each treatment was taken hourly for determination of the viable cell counts, diluted (1:10, *v*/*v*) with 0.1% (*w*/*v*) sterile buffered peptone water (Amyl Media Pty. Ltd., Dandenong, Australia), and mixed uniformly with a vortex mixer. Serial dilutions were prepared and viable numbers enumerated using spread plating on MRS-salicin agar and colonies were counted after 48 h incubation anaerobically at 37 °C as described by Guan (2017) [15]. To determine bile salt tolerance, the free and freeze-dried cells were inoculated into a milk yeast extract medium at pH 6.0 with 0 (control), 5, 10 g/L salts (Oxgall, Sigma). The sampling was carried out in triplicate for 3 h of incubation at 37 °C; and enumeration of free and freeze-dried cells was determined on MRS agar as described above in the preceding section. Duplicate samples were withdrawn after incubation at 37 °C for 0, 3, and 6 h and cell counts of free, encapsulated, and co-encapsulated bacteria were enumerated on MRS-salicin agar as described previously. To determine the viable counts of the encapsulated bacteria, test sample contents were centrifuged (3000× *g* for 10 min at 4 °C) and the capsules dissolved by re-suspending in 9.0 mL of phosphate buffer (0.1 M, pH 7.0) followed by gentle shaking at room temperature for 15 min. The number of released cells was determined by spread plate count using MRS-salicin agar as described previously [16,17].

### 2.3. Particle Size and Moisture Content

The particle size of the microcapsules was measured by scanning electron microscopy (SEM) software at 10.0 kV (Hitachi S-3400, Krefeld, Germany) and examined by the light-scattering method by a Mastersizer 2000 (Malvern, Worcestershire, UK) equipped with a humid sampling unit (Hydro 2000S). Isopropanol was used as a dispersing medium for the samples. The refractive index was set to 1.52. The particle size of the microcapsules was evaluated as both surface mean diameter (D3, 2) and volume mean diameter (D4, 3). The moisture content was determined with a moisture analyzer (Kern DBS, Balingen, Germany) at 105 °C [18,19].

### 2.4. Scanning Electron Microscopy (SEM)

The samples were cross-sectioned and mounted on slabs with adhesive tape. The morphology of the specimens were dried to critical point, coated with gold and investigated by Scanning Electron Microscopy (SEM) at 10.0 kV (Hitachi S-3400, Krefeld, Germany). The dimensions of the capsules were determined using an objective micrometer on an optical microscope at a 400× magnification. The dispersion of the cells in the arrowroot matrix was examined by staining with iodine and observing under light microscopy. The 35 mm diameter cylindrical aluminum probe at a speed of 0.1 mm/s in a compression mode, and a rupture distance of 1.0 mm, was used. The peak force was measured in grams. The beads were tested each time and 3 replications were applied for each treatment [11].

### 2.5. Microencapsulation and Coating Procedures

The extrusion technique of microencapsulation was derived from Kailasapathy et al., (2002) by using arrowroot as the supporting matrix. To form beads, the arrowroot solution was extruded into a previous sterile whey solution and stirred. The beads were sieved off from the solution and washed with sterile distilled water.

Wall materials were composed of arrowroot starch, maltodextrin, and whey protein for the microencapsulation of the bacteria. Chemical reagents of analytical grade were used for various characteristic tests of encapsulated bacteria. An aqueous phase was prepared by dissolving whey protein and maltodextrin or arrowroot starch (maltodextrin or arrowroot starch mix with whey protein as described in Table 1) in deionized water at ice batch using a homogenizer at 10,000 rpm for 5 min. When total solution was obtained, bacteria were added gently to the solution. The mixture was emulsified using a homogenizer at 10,000 rpm for 6 min and 500 psi (MFG Company, Chicago, IL, USA) [10,20,21]. Finally, the samples were freeze-dried in a lyophilizer (Freeze Dryer Lyobeta 25) under vacuum at −40 °C for 20 h. Freeze-dried bacteria samples were immediately transferred to sample bottles, which were sealed and stored in at −20 °C [22,23].

### 2.6. Effects of Bacteria Encapsulated Arrowroot and Maltodextrin on Improving the Survival of Probiotics Organisms in Yoghurt

The incorporation of 2% prebiotics from arrowroot starch and maltodextrin (Laboratory of Food Biochemistry, NPUST, Pingtung, Taiwan), was carried out in skimmed milk samples and yogurt made from *L. plantarum*, *W. paramesenteroides*, *E. faecalis*, and *L. paraplantarum*. The bacteria counts were performed after storage for periods of 0, 15, 30, 60, and 90 days at 4 °C and 25 °C in yogurt samples in both encapsulated bacteria. To determine the viable counts of the bacteria, test sample contents were centrifuged (3000× *g* for 10 min at 4 °C) and the yogurt were dissolved by re-suspending in 9.0 mL of phosphate buffer (0.1 M, pH 7.0) followed by gentle shaking at room temperature for 15 min. The number of released cells by dilution was determined by spread plate count using MRS-salicin agar as described previously [26,27].

### 2.7. Statistical Analysis

Each assay was repeated on three independent occasions with triplicate determinations. Statistical analysis was performed using the Pearson product moment correlation coefficient and SPSS 13.0 software (SPSS, Inc., Chicago, IL, USA) with statistical significance determined at *p* ≤ 0.05. Results are expressed as the mean and standard error of the mean of three independent experiments. One-way analysis of variance followed by least significant difference test was used to determine significant differences of viability of the tested strains in simulated gastrointestinal fluid.

## 3. Results and Discussion

### 3.1. Survival of Probiotic Strains in pH and Bile

The pH remains a crucial parameter in the viability of probiotic bacterial strains, and it is observed that the viability of microorganisms is limited to an acidic medium. However, in this report, there was no significant decrease in active numbers of freeze-dried probiotic bacteria (Table 2). The strains *W. paramesenteroides* was the most constant in count at pH 2.0 (log 7.89 cfu/mL) and pH 3.0 (log 7.74 cfu/mL) respectively, whereas in free cells, it was about log 8.97 cfu/mL at pH 2 and log 8.83 cfu/mL at pH 3. Similarly, *E. faecium* survived at pH 2 (log 7.77 cfu/mL) and pH 3 (log 8.21 cfu/mL) for the freeze-dried cells while *L. plantarum* exhibited the lowest count at pH 2 and 3. All the freeze-dried and free cell samples were stable at pH 6 where all the strains showed counts above log 6 cfu/mL which is the human daily demand in probiotics. All the bacteria showed strong bile salt counts for both free and freeze-dried cells at a concentration of 0.3%, which is referential for the human stomach. The viability of those organisms was assessed at different concentration of bile salt 0.3%, 0.5%, and 0.8% (Table 2) where the 4 strains exhibited good responses to increases in salt concentrations. The freeze-dried bacteria *E faecalis* showed very strong counts at different bile salt concentrations of 0.5% (log 9.21 cfu/mL), 0.3% (log 7.91 cfu/mL), and 0.8% (log 7.84 cfu/mL); therefore, it was greater than similar bile salt concentrations for the free cells. Freeze-dried *L. plantarum* exhibited the lowest bile salt concentrations comparing to all the strains, but still remained above the log 6 cfu/mL.

### 3.2. Survival of Encapsulated Bacteria in Simulated Intestinal Juice at 4 °C and 25 °C

Table 3 summarized the probability of the viability of the arrowroot- and maltodextrin- microencapsulated probiotic bacteria in a simulated GIT. After the 90 days’ exposure to simulated gastrointestinal tract for both maltodextrin and arrowroot encapsulated *L. paraplantarum*, *E. faecalis*, *L. plantarum* and *W paramsenteroides*, their viability was 68.2%, 73.63%, 77.34%, and 73.73% versus 69.45%, 79.24%, 77.85%, and 72.77%, respectively, of their initial population at 4 °C. Meanwhile, microencapsulated cells were also resistant to simulated gastric conditions after 90 days with a viability of 72.64%, 69.45%, 74.92%, and 73.20% versus 71.79%, 69.57%, 69.92%, and 71.37% of the initial population found in arrowroot and maltodextrin coated *L. paraplantarum*, *E. faecalis*, *L. plantarum*, and *W paramsenteroides*, respectively at 25 °C (Table 2). Encapsulated maltodextrin *L. paraplantarum*, *E. faecalis*, *L. plantarum*, and *W. paramesenteroides* showed great viability after 0 days corresponding to 8.40, 8.76, 8.65, and 8.87 log cfu/mL, following a significant decrease after 90 days of 5.73, 6.45, 6.69, and 6.54 log cfu/mL at 4 °C; whereas, the counts were estimated to 8.37, 8.61, 9.05, and 8.73 log cfu/mL with a slight decrease after 90 days of 6.08, 5.98, 6.78, and 6.39 log cfu/mL respectively at 25 °C at *p* < 0.05 (Table 2). Furthermore, arrowroot coated *L. paraplantarum*, *E. faecalis*, *L. plantarum*, and *W. paramesenteroides* exhibited counts of 8.74, 8.43, 8.76, and 9.11 log cfu/mL with a significant decrease compared to 90 days of 6.07, 6.68, 6.82, and 6.63 log cfu/mL at 4 °C; whereas, it was estimated to be 8.40, 8.84, 8.71, and 9.08 log cfu/mL, while after 90 days to be 6.03, 6.15, 6.09, and 6.48 log cfu/mL, respectively, at 25 °C *p* < 0.05. Furthermore, a quick decrease of bacterial cells was observed in yogurt after 90 days; initial counts of 10^9^ cfu/mL for bacteria quickly dropped to less than 10^3^ cfu/mL after exposure, but in only 15 days microcapsule bacteria decreased to more than 10 cfu/mL.

### 3.3. Moisture Content of the Microcapsules

Table 3 shows the water concentration of the maltodextrin-coated probiotic cells at storage. *E. faecalis* exhibited 5.09% moisture content, which was the lowest value for maltodextrin capsules. Under the encapsulation conditions tested, the probiotic *W. paramesenteroides* showed the best moisture content value of 5.47% in comparison to other maltodextrin-encapsulated bacteria. As expected, the arrowroot encapsulated bacteria showed higher moisture content for all the bacteria compared to the maltodextrin capsules. For example, most of the arrowroot capsules were around 8% moisture content; where *E. faecalis* showed the lowest moisture content value of 8.01% at 4 °C and 8.09% at 25 °C; while *L. plantarum* exhibited 8.11% at 4 °C and 8.20% at 25 °C; *L. paraplantarum*, 8.31% at 4 °C and 8.39% at 25 °C; and *W. paramesenteroides*, 8.17% at 4 °C and 8.23% at 25 °C (Table 3). The moisture content was increased in the arrowroot capsules at both room and refrigerated conditions. Arslan (2017) found similar findings in his report where the moisture content for probiotic capsules was around 11% for freeze-drying and 4.41% for spray-chilling methods [22].

### 3.4. Correlation Coefficients of Cells and Capsules

All the isolates of *L. paraplantarum*, *W. paramesenteroides*, *E. faecalis*, and *L. plantarum* indicated a certain correlation performing different methods of encapsulation, while free cells and freeze-dried capsules showed less significant correlation at p-values 0.01 and 0.05. Furthermore, the results of this research showed a significant correlation between the different arrowroot encapsulated bacteria and freeze-dried cells (Table 4 and Table 5). However, free-cells and freeze-dried bacteria were not significantly correlated with the control at *p*-value 0.01 and 0.05 (Table 5); free *L. plantarum* cells showed significant correlation with all the freeze-dried isolates at *p*-values 0.01 and 0.05, respectively. Nonetheless, a different relationship was observed for free *L. paraplantarum* that showed no correlation with the freeze-dried isolates. Free *W. paramesenteroides* cells showed significant correlation among most of the freeze-dried isolates at *p*-values 0.01 and 0.05 (Table 5); moreover, free *E. faecalis*, *W. paramesenteroides*, and *L. plantarum* reported significant correlation with all the freeze-dried strains at different *p*-values (*p* = 0.01 and *p* = 0.05). The correlation between the arrowroot and maltodextrin microcapsules were more significant compared to the free cells and freeze-dried bacteria. Most of the cells showed significant correlation to each other and with the control at *p*-values 0.01 and 0.05. Encapsulated arrowroot *L. paraplantarum* indicated better correlation with all the encapsulated maltodextrin strains at *p* = 0.01 while encapsulated arrowroot *L. plantarum* significantly correlated with maltodextrin microcapsules at *p* = 0.05.

### 3.5. Survival of Strains in pH and Bile

The stereo microscope images showed freeze-dried bacteria were rod-shaped for most of the *L. plantarum*, *L. paraplantarum*, and *E. faecalis*, *whereas W. paramesenteroides* showed a bacilli shape (Figure 1). However, the arrowroot microcapsules and maltodextrin capsules were considered to be smaller and showed an opaque aspect with a crystalline form for the maltodextrin (Figure 2). The scanning electron microscope photos of microscopic beads (Figure 2) showed that freeze-dried bacteria are closer to the microcapsule form of arrowroot and maltodextrin media.

Figure 1 and Figure 2 illustrated the diameters and coating forms of arrowroot and maltodextrin microscopic spheres with or without whey incorporating *Lactobacilli* cells. The average diameter of freeze-dried *L. plantarum*, *W. paramesenteroides*, *E. faecalis*, and *L. paraplantarum* microspheres was 1.22 μm after storage. *W. paramesenteroides* showed higher diameter of 1.47 μm and *L. paraplantarum* showed a lower diameter of 997.03 nm. The mean diameters of probiotic bacteria-encapsulated arrowroot microspheres *L. plantarum, W. paramesenteroides, E. faecalis,* and *L. paraplantarum* were 382.8 μm and maltodextrin-coated bacteria were between 349.92 and 458.91 μm (Table 2). To determine the size of the capsules, electronic microphotographs were performed using optical microscope. Thus, the sizes of the capsules varied based on the probiotic isolates and the material used in encapsulation. In this research, the microencapsulation technique that was adopted increased the sizes of the capsules from 0.5 to 1 mm. Thus, sieves of 1 mm were used to separate the different sizes. Different forms of capsules were observed as spherical forms or elliptical forms. Starch grains were observed throughout all concavities, similar to those described above, although the distribution of bacteria was random in the arrowroot and maltodextrin matrices (Figure 1 and Figure 2).

## 4. Discussion

A controlled and randomized study was conducted on 53 healthy volunteers consuming for 3 weeks a drink of 3 lyophilized probiotic organisms (*L. fermentum* ME-3, *L. paracasei* 8700:2 and *B. longum* 46, 6 × 10^9^ cfu/day) associated to a prebiotic (Raftilose) or (maltodextrin) [28]. Among the volunteers, some were colonized by *Helicobacter pyolori*. In the latter, the consumption of the symbiotic significantly reduced the redox status and significantly increased the total antioxidant status compared to maltodextrin [29]. The survival of the microcapsules was in agreement with other related researches, which reported the benefits of encapsulation and the high viability among different pH and bile conditions. Similarly to our findings, Sultana et al., (2000) found no significant increase in the viability of encapsulated bacteria after exposure to low pH and bile conditions [10,30,31,32]. Trindade et al., (2000) reported that *B. bifidum* and *L. acidophilus* encapsulated in calcium alginate capsules showed no significant counts in maintaining cells from 2% to 4% bile salt [33]. In this report, symbiosis had a strong effectiveness on the viability of the organisms at a higher level, due to the fermentation of carbohydrates by bacteria and production of short chain fatty acids and gases. Those fermented byproducts are considered to be healthy for the hosts.

Bile is a water-soluble cholesterol product in the liver that is massed in the gallbladder to be freed into the duodenum when food is ingested [34]. Generally, prebiotics form a symbiotic relationship with bacteria that sometimes help to reduce the cholesterolaemia through certain mechanisms. One of the mechanisms of this symbiosis is the reduction of cholesterol absorption followed by an increased excretion of feces; on the other hand, fermentation allows bacterial microflora to produce short chain fatty acids in the intestines [35]. Furthermore, prebiotics associated with bacteria reduce the production of bile; and, in turn, the production of bile salts hydrolase (BSH) will be stimulated. Thus, conjugated bile acids of glycol and tauro-deoxycholic will be hydrolyzed by BSH leading to the deconjugation of bile acids [36].

It is reported that capsules of bacterial probiotics were more suitable to preserve survivability in acidic simulated GIT [30]. The coating of bacterial probiotics in alginate capsules were analyzed to improve the survivability of probiotic cells in simulated GIT [37,38]. Furthermore, Chávarri et al. (2010) highlighted the viability levels of bifidobacteria in alginate capsules incorporated with chitosan were greater than those of alginate capsules [39]. However, Sultana et al. (2000) noticed that the microencapsulation of cells in alginate capsules failed to effectively protect the cells from high pH conditions [10]. Therefore, some researchers highlighted the incidence of poor or lack of uniformity of alginate encapsulation on viability of LAB in simulated GIT, and the coating procedure [10,11,40]. Various studies have exhibited gaps amongst isolates of probiotic cells in agreement with their viability in low pH condition [5,6]. In particular, it is indicated that the Bifidobacterium isolates constituted the most susceptible acidic bacteria, which was evaluated in this research [41]. Arrowroot-encapsulated strains exhibited higher active counts than maltodextrin-encapsulated strains, which was expected due to their greater resistance to low pH. The added encapsulation allowed greater protection to bacterial cells in comparison to free cells at similar times.

Encapsulated arrowroot and maltodextrin bacteria generated hydrophobically showed a regular and identical aspect, while freeze-dried bacteria showed a certain space from one bacteria to another. These findings could be related to the cooling of the hydrophobic material that occurs due to their low heat transfer to dry the beads. Freeze-dried beads had both a round and bacilli shape, soft surface and concavities; however, the capsules showed only a round shape inside the different materials of arrowroot and maltodextrin that indicated the fundamental origin of the concavities was a drying feature of the membrane. These structures may be established by the fast drying process, which led to crustiness and reduced diffusion of the water on top of the beads; in addition, there was a high-pressure established surface [42,43].

In a study conducted on microcapsule of *B. longum and B. infantis*, Lian et al., (2002) indicated the effect of different wall materials for bacteria with the greatest moisture content of 8.61 to 10.31% in gum Arabic [44]. Overall, bacteria live better in a low moisture content environment; therefore, it remains a crucial factor in the reliability of dried cells. However, excessive drying might decrease the survival and reliability of bacterial cells. In addition, variations in water content depend on the composition of the liquid wherein the bacteria cells were dried, on storage, and on the organisms [45].

## 5. Conclusions

The study showed that the prebiotic arrowroot and maltodextrin composite encapsulation improved the viability of bacteria under low pH conditions, and stomach simulated bile salts. Bacteria-encapsulated arrowroot showed greater viability than bacteria-encapsulated maltodextrin and freeze-drying techniques. Therefore, arrowroot-encapsulated microcapsules could be used to administer active probiotic bacteria to the simulated GIT. In conclusion, the microencapsulation of *L. plantarum*, *W. paramesenteroides*, *E. faecalis* and *L. paraplantarum* with a coating of arrowroot and maltodextrin proved an effective way of delivering live bacteria at suitable levels to the intestines and served to maintain their viability in the simulated GIT. Moreover, microencapsulation increased the survivability of bacterial cells in yoghurt after 90 days’ storage at room and refrigerated conditions. Freeze-drying and microencapsulation using arrowroot and maltodextrin composite materials presented an innovative technique for increasing the survival of LAB.

## Figures and Tables

**Figure 1 foods-08-00175-f001:**
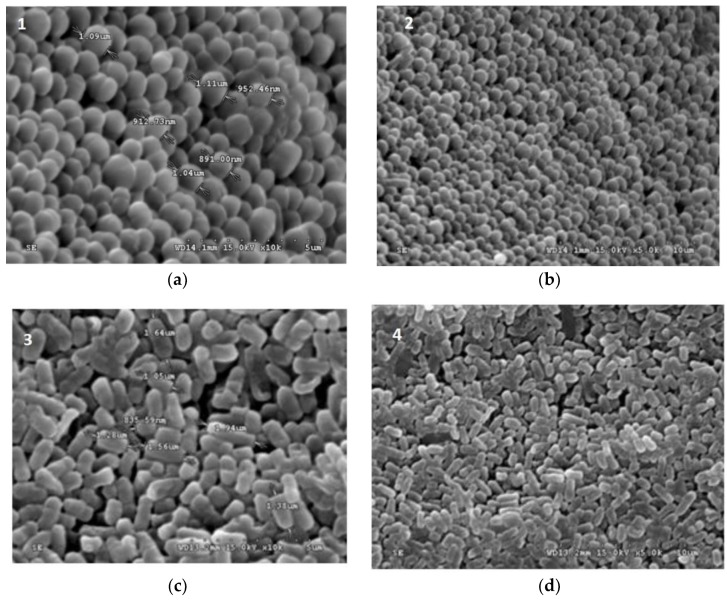
Scanning electron micrograph (SEM) of freeze dried (**a**) *L. plantarum*, (**b**) *L. paraplantarum*, (**c**) *W. paramesenteroides*, and (**d**) *E. faecalis*.

**Figure 2 foods-08-00175-f002:**
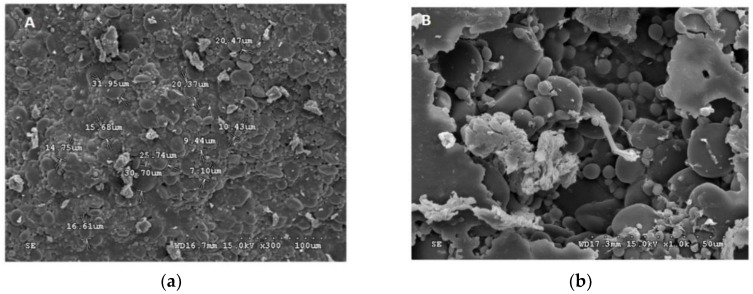
Scanning electron micrograph (SEM) of the encapsulated beads. (**a**) *W. paramesenteroides*-encapsulated arrowroot, (**b**) *W. paramesenteroides*-encapsulated maltodextrin, (**c**) *L. paraplantarum*-encapsulated maltodextrin, (**d**) *L. paraplantarum*-encapsulated arrowroot, (**e**) *E faecalis*-encapsulated arrowroot, (**f**) *E faecalis*-encapsulated maltodextrin, (**g**) *L. plantarum*-encapsulated arrowroot, and (**h**) *L. plantarum*-encapsulated maltodextrin.

**Table 1 foods-08-00175-t001:** Schema of encapsulated bacteria formulation [24,25].

Formulation Code	Arrowroot Starch (g)	Wall Materials Combination		Strains (g)	Water (mL)
		Maltodextrin (g)	Whey Protein (g)		
Smb_1_	-	0.625	0.125	0.25	1
Smb_2_	-	0.650	0.10	0.25	1
Sab_1_	0.625	-	0.125	0.25	1
Sab_2_	0.650	-	0.10	0.25	1

Smb: Symbiotic maltodextrin bacteria and Sab: Symbiotic arrowroot bacteria.

**Table 2 foods-08-00175-t002:** Comparison of free and freeze-dried bacteria counts in simulated intestinal juices and bile salt conditions after incubation.

Treatments	Isolates Name	Initial Mean Counts	Simulated Intestinal Juices			Bile Salt		
			pH 2	pH 3	pH 6	0.3%	0.5%	0.8%
	*L. paraplantarum*	8.84 ± 0.23 a ^2^	7.39 ± 0.32 b	8.26 ± 0.22 b	8.73 ± 0.17 a	7.24 ± 0.34 b	8.84 ± 0.23 a	6.81 ± 0.37 b
Free-cells	*E. faecalis*	9.13 ± 0.19 a	7.77 ± 0.28 b	8.21 ± 0.29 b	9.19 ± 0.12 a	7.79 ± 0.27 b	9.13 ± 0.19 a	7.73 ± 0.41 b
	*L. plantarum*	8.92 ± 0.13 a	7.13 ± 0.36 b	8.58 ± 0.36 b	8.81 ± 0.18 a	6.84 ± 0.42 b	8.92 ± 0.13 a	6.64 ± 0.29 c
	*W. paramesenteroides*	9.03 ± 0.25 a	7.89 ± 0.22 c	7.74 ± 0.33 b	9.08 ± 0.16 a	6.51 ± 0.25 b	9.03 ± 0.25 a	7.18 ± 0.38 c
	*L. paraplantarum*	8.53 ± 0.16 a	7.83 ± 0.37 c	7.59 ± 0.21 b	8.84 ± 0.13 a	7.78 ± 0.38 b	8.93 ± 0.16 a	6.49 ± 0.39 b
Freeze-dried	*E. faecalis*	9.01 ± 0.29 a	8.15 ± 0.28 b	8.41 ± 0.29 a	9.21 ± 0.29 a	7.91 ± 0.35 a	9.21 ± 0.26 a	7.84 ± 0.13 a
	*L. plantarum*	9.11 ± 0.19 a	8.43 ± 0.36 b	8.87 ± 0.32 a	9.19 ± 0.31 a	9.01 ± 0.23 a	8.91 ± 0.15 a	9.12 ± 0.32 a
	*W. paramesenteroides*	8.93 ± 0.21 a	8.97 ± 0.32 c	8.83 ± 0.34 a	8.97 ± 0.09 a	7.43 ± 0.27 a	8.94 ± 0.32 a	8.97 ± 0.19 a

Each value represents the mean value–standard deviation (SD) from three trials. ^2^ Any two means in the same column having different letters represent significant difference at *p* < 0.05.

**Table 3 foods-08-00175-t003:** Viability of encapsulated arrowroot and maltodextrin combined probiotics *L. paraplantarum*, *L. plantarum*, *W. paramesenteroides*, and *E. faecalis* (log10 cfu/mL) ^1^ in yoghurt during storage for 3 months at 4 and 25 °C.

Temperature	Treatment	Microsphere Size (μm; *n* = 100)	Moisture Content (%)	Log10 cfu/mL					Survival (%)
				0 Day	15 Days	30 Days	60 Days	90 Days	
	*L. paraplantarum*	363.09 ± 3.16 a	5.27 ± 0.69 a	8.40 ± 0.19 a	7.91 ± 0.62 a	6.65 ± 0.12 c	6.51 ± 0.21 c	5.73 ± 0.23 d	68.2 ± 0.73 a
4 °C (Maltodextrin-coated)	*E. faecalis*	426.17 ± 6.38 b	5.09 ± 0.45 a	8.76 ± 0.31 a	8.31 ± 0.35 a	7.36 ± 0.36 b	7.58 ± 0.13 a	6.45 ± 0.38 d	73.63 ± 1.28 a
	*L. plantarum*	354.11 ± 2.96 a	5.31 ± 0.28 a	8.65 ± 0.32 a	8.43 ± 0.18 a	7.21 ± 0.44 b	7.73 ± 0.35 a	6.69 ± 0.22 c	77.34 ± 0.81 a
	*W. paramesenteroides*	458.91 ± 6.29 b	5.42 ± 0.17 a	8.87 ±0.28 a	9.03 ± 0.29 a	7.28 ± 0.72 b	6.42 ± 0.42 c	6.54 ± 0.48 d	73.73 ± 1.64 a
	*L. paraplantarum*	349.92 ± 3.09 a	5.36 ± 0.83 a	8.37 ± 0.32 a	8.09 ± 0.57 a	7.53 ± 0.29 a	6.47 ± 0.27 c	6.08 ± 0.18 d	72.64 ± 0.62 a
25 °C (Maltodextrin-coated)	*E. faecalis*	418.64 ± 6.05 a	5.19 ± 0.96 a	8.61 ± 0.11 a	8.18 ± 0.48 a	6.96 ± 0.48 b	7.42 ± 0.08 a	5.98 ± 0.83 d	69.45 ± 0.92 a
	*L. plantarum*	318.19 ± 2.89 a	5.42 ± 0.31 a	9.05 ± 0.42 a	8.33 ± 0.19 a	7.42 ± 0.79 a	7.68 ± 0.74 a	6.78 ± 0.67 c	74.92 ± 0.68 a
	*W. paramesenteroides*	442.37 ± 6.15 b	5.47 ± 0.53 a	8.73 ± 0.92 a	8.84 ± 0.82 a	8.04 ± 0.82 a	7.37 ± 0.19 b	6.39 ± 0.85 d	73.20 ± 0.98 a
	*L. paraplantarum*	571.09 ± 3.41 c	8.31 ± 0.47 b	8.74 ± 0.83 a	8.02 ± 0.74 a	6.95 ± 0.09 c	6.67 ± 0.89 c	6.07 ± 0.93 d	69.45 ± 0.25 a
4 °C (Arrowroot starch-coated)	*E. faecalis*	643.18 ± 5.49 d	8.01 ± 0.78 b	8.43 ± 0.57 a	8.29 ± 0.93 a	7.57 ± 0.67 a	7.85 ± 0.78 a	6.68 ± 0.65 c	79.24 ± 0.92 a
	*L. plantarum*	537.08 ± 2.89 c	8.11 ± 0.35 b	8.76 ± 0.29 a	8.56 ± 0.84 a	7.51 ± 0.94 b	7.86 ± 0.81 a	6.82 ± 0.28 c	77.85 ± 0.82 a
	*W. paramesenteroides*	681.34 ± 5.75 d	8.17 ± 0.19 b	9.11 ± 0.71 a	9.07 ± 0.27 a	8.13 ± 0.78 a	6.94 ± 0.65 b	6.63 ± 0.92 c	72.77 ± 0.62 a
	*L. paraplantarum*	527.09 ± 2.78 c	8.39 ± 0.21 b	8.40 ± 0.78 a	7.82 ± 0.68 a	7.58 ± 0.85 a	6.57 ± 0.81 c	6.03 ± 0.48 d	71.79 ± 0.62 a
25 °C (Arrowroot starch-coated)	*E. faecalis*	637.43 ± 5.83 d	8.09 ± 0.18 b	8.84 ± 0.28 a	8.47 ± 0.58 a	7.19 ± 0.58 b	7.59 ± 0.47 a	6.15 ± 0.72 d	69.57 ± 1.36 a
	*L. plantarum*	523.26 ± 3.18 c	8.20 ± 0.23 b	8.71 ± 0.49 a	8.72 ± 0.72 a	7.63 ± 0.83 b	7.77 ± 0.84 a	6.09 ± 0.28 d	69.92 ± 1.12 a
	*W. paramesenteroides*	668.37 ± 3.11 d	8.23 ± 0.28 b	9.08 ± 0.69 a	8.93 ± 0.37 a	8.09 ± 0.91 a	7.54 ± 0.38 a	6.48 ± 0.84 d	71.37 ± 0.23 a

^1^ Means ± standard deviation (SD). Means with different letters within the same row indicate significant difference at *p* < 0.05.

**Table 4 foods-08-00175-t004:** Pearson’s correlation coefficients between the encapsulated maltodextrin and arrowroot bacteria.

		Control	*L. plantarum*	*W. paramesenteroides*	*E. Faecalis*	*L. paraplantarum*	Control	*L. Plantarum*	*W. paramesenteroides*	*E. Faecalis*	*L. paraplantarum*
	Control	1									
	*L. plantarum*	0.898 *	1								
**Maltodextrin**	*W. paramesenteroides*	0.666	0.896 *	1							
	*E. faecalis*	0.842	0.810	0.528	1						
	*L. paraplantarum*	0.892 *	0.997 **	0.919 *	0.770	1					
	Control	0.993 **	0.907 *	0.653	0.896 *	0.892 *	1				
	*L. plantarum*	0.813	0.940 *	0.969 **	0.637	0.963 **	0.792	1			
**Arrowroot**	*W. paramesenteroides*	0.783	0.823	0.881 *	0.468	0.864	0.728	0.948 *	1		
	*E. Faecalis*	0.862	0.989 **	0.910 *	0.731	0.990 **	0.865	0.934 *	0.823	1	
	*L. paraplantarum*	0.927 *	0.971 **	0.873	0.712	0.980 **	0.909 *	0.942 *	0.895 *	0.975 **	1

**: means the values are significantly correlated at *p*-value = 0.01; *: means the values are significantly correlated at *p*-value = 0.05.

**Table 5 foods-08-00175-t005:** Pearson’s correlation coefficients between the capsules of freeze-dried and the free cells.

		Control	*L. plantarum*	*W. paramesenteroides*	*E. Faecalis*	*L. paraplantarum*	Control	*L. Plantarum*	*W. paramesenteroides*	*E. Faecalis*	*L. paraplantarum*
**Free-cells**	Control	1									
	*L. plantarum*	0.620	1								
	*W. paramesenteroides*	0.361	0.688	1							
	*E. faecalis*	0.626	0.960 **	0.617	1						
	*L. paraplantarum*	0.800	0.502	0.590	0.602	1					
**Capsules of Freeze-dried**	Control	0.503	0.947 *	0.443	0.933 *	0.303	1				
	*L. plantarum*	0.768	0.933 *	0.799	0.863	0.675	0.778	1			
	*W. paramesenteroides*	0.583	0.958 *	0.811	0.855	0.474	0.837	0.962 **	1		
	*E. Faecalis*	0.868	0.841	0.639	0.740	0.630	0.696	0.954 *	0.882 *	1	
	*L. paraplantarum*	0.791	0.946 *	0.622	0.866	0.549	0.862	0.963 **	0.938 *	0.964 **	1

**: means the values are significantly correlated at *p*-value = 0.01; *: means the values are significantly correlated at *p*-value = 0.05.

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
