# Peer review of "Viability of 4 Probiotic Bacteria Microencapsulated with Arrowroot Starch in the Simulated Gastrointestinal Tract (GIT) and Yoghurt"

_foods, 2019, doi:10.3390/foods8050175_

Reviewer 1 Report

Introduction:

Comments: The following information needs to be included for a more coherent for examples, the GIT response of stress as it impact the survival rate of the probiotic. Provide some literature to justify your research. In line 118, seem arrowroot starch and maltodextrin are used as prebiotics, hence there was no mention of this materials in the introduction.  Provide some information to justify the used of this materials and how it may influence the use of the probiotics and its impact on the GIT. The application of a homogenizer for microencapsulation and coating is not clear, please include some information on the principles of microencapsulation using a homogenizers.

Line 83 to 92: Provide details of the type of system use for the simulation study as there are indications that the pH was controlled how about the temperature. Define what is meant by Free cells

Line 93: The mastersizer does to measure the actual particle diameter, but the hydrodynamic diameter, please adjust accordingly.  It will be interesting to include polydispersity index and a plot of the size distribution.  What is (D3, 2).

Line 100 to 103: Please provide more details of the procedure, level of magnifications etc.

Line 108/109: Include the specification of the homogenizers used. What is the pressure of the homogenizer? Does it have any effect on the bacteria survival?

Line 118-122 is not clear. Provide details on how bacterial counts were conducted. A 0, 15, 30, 60 and 90 days At 4 and 25°C shelflife study was conducted but justification was given in the introduction. Please include some justification in the introduction. 

Line 171: Is this the moisture content of the microcapsules or the bacteria

Line 190: How was correlation determined, please include this methods under the statistical analysis on line 123.

Comments: Please reorder the presentation of the materials and methods based on the following suggestions: Subheading 2.1, 2.5, 2.3, 2.4, 2.6, 2.2

Line 297. How was the stability of the bacterial determined?

Author Response

Open Review

Comments and Suggestions for Authors

Introduction:

 Point 1: Comments: The following information needs to be included for a more coherent for examples, the GIT response of stress as it impact the survival rate of the probiotic. Provide some literature to justify your research. In line 118, seem arrowroot starch and maltodextrin are used as prebiotics, hence there was no mention of this materials in the introduction.  Provide some information to justify the used of this materials and how it may influence the use of the probiotics and its impact on the GIT. The application of a homogenizer for microencapsulation and coating is not clear, please include some information on the principles of microencapsulation using a homogenizers.

Response 1: Sultana reported that the incorporation of Hi‐Maize® starch improved encapsulation of Lactobacillus acidophilus and Bifidobacterium spp. in yoghurt, as compared to when the bacteria were encapsulated without the starch [10]. Habitually, 1–2% insoluble starch grains added to the probiotic–hydrocolloid precursor directly before the encapsulation process, aim to further maintain the viability of probiotics [10-11]. Moreover, arrowroot starch, a commercially underexploited tuber starch but having potential digestive and medicinal properties, is a natural healer. It has a percentage digestibility of 30.07% after 30 min of incubation with the enzyme and 25.27% to 30.56% after extrusion. Its absorption index is 6.52 to 8.85 g gel/g dry sample, water solubility index 15.92% to 41.31%, and oil absorption index 0.50 to 1.70 g/g were higher for the extrudates comparing to native starch of 1.81 g gel/g dry sample, 1.16% and 0.60 g/g, respectively [12].

A homogenizer at 3000 rpm for 5 minutes. When total solution was obtained, bacteria were added gently to the solution. The mixture was emulsified using a homogenizer at 10000 rpm for 6 minutes and 500 psi (MFG Company, Chicago, IL)[10, 21-22]. Finally, the samples were freeze-dried in a lyophilizer (Freeze Dryer Lyobeta 25) under vacuum at -40 °C for 20 hours.

Point 2: Line 83 to 92: Provide details of the type of system use for the simulation study as there are indications that the pH was controlled how about the temperature. Define what is meant by Free-cells

Response 2: The encapsulated and free bacteria were added to Non-fat milk, glucose, yeast extract and cysteine medium (12% non-fat skim milk, 2.0% glucose, 1.0% yeast extract, and 0.05% cysteine) (NGYC) that had been adjusted to pH 2.0, pH 3.0, or 6.5 (control) with 5 M HCl or 1 M NaOH in 10 ml aliquots. The samples were incubated at 37 °C for 3 h. An aliquot from each treatment was taken hourly for determination of the viable cell counts, diluted (1:10, v/v) with 0.1% (w/v) sterile buffered peptone water (Amyl Media Pty. Ltd., Dandenong, Australia), and mixed uniformly with a vortex mixer. Serial dilutions were prepared and viable numbers enumerated using spread plating on MRS-salicin agar and colonies counted after 48 h incubation anaerobically at 37 °C as described by Guan (2017)[15]. To determine bile salt tolerance, the free and freeze-dried cells were inoculated into a milk yeast extract medium at pH 6.0 with 0 (control), 5, 10 g/l salts (Oxgall, Sigma). The sampling was carried out in triplicate for 3 hours of incubation at 37 oC; and enumeration of free and freeze-dried cells was determined on MRS agar as described above in the preceding section. Duplicate samples were withdrawn after incubation at 37 °C for 0, 3, and 6 h and cell counts of free, encapsulated, and co-encapsulated bacteria were enumerated on MRS-salicin agar as described previously. To determine the viable counts of the encapsulated bacteria, test sample contents were centrifuged (3000 g for 10 min at 4 °C) and the capsules dissolved by re-suspending in 9.0 mL of phosphate buffer (0.1 M, pH 7.0) followed by gentle shaking at room temperature for 15 min. The number of released cells was determined by spread plate count using MRS-salicin agar as described previously [16-17].

Point 3: Line 93: The mastersizer does to measure the actual particle diameter, but the hydrodynamic diameter, please adjust accordingly.  It will be interesting to include polydispersity index and a plot of the size distribution.  What is (D3, 2).

Response 3: The particle size of the microcapsules was measured by the software of the Scanning Electron Microscopy (SEM) at 10.0 kV (Hitachi S-3400, Germany) and examined by the light scattering method by a Mastersizer 2000 (Malvern, Worcestershire, UK) equipped with a humid sampling unit (Hydro 2000S). Isopropanol was used as a dispersing medium for the samples. The refractive index was set to 1.52. The particle size of the microcapsules was evaluated as both surface mean diameter (D3, 2) and volume mean diameter (D4, 3).

In fluid dynamics, Sauter mean diameter (SMD, d32 or D[3, 2]) is an average of particle size. It was originally developed by German scientist Josef Sauter in the late 1920s. It is defined as the diameter of a sphere that has the same volume/surface area ratio as a particle of interest.

Point 4: Line 100 to 103: Please provide more details of the procedure, level of magnifications etc.

Response 4: The dimensions of the capsules were determined using an objective micrometer on an optical microscope at a 400x magnification. The dispersion of the cells in the arrowroot matrix was examined by staining with iodine and observing under light microscopy. The 35 mm diameter cylindrical aluminum probe at a speed of 0.1 mm/s in a compression mode, and the rupture distance of 1.0 mm was used. The peak force was measured in grams.

Point 5: Line 108/109: Include the specification of the homogenizers used. What is the pressure of the homogenizer? Does it have any effect on the bacteria survival?

Response 5: The morphology of the specimens were dried to critical point, coated with gold and investigated by Scanning Electron Microscopy (SEM) at 10.0 kV (Hitachi S-3400, Germany). The dimensions of the capsules were determined using an objective micrometer on an optical microscope at a 400x magnification. The dispersion of the cells in the arrowroot matrix was examined by staining with iodine and observing under light microscopy. The 35 mm diameter cylindrical aluminum probe at a speed of 0.1 mm/s in a compression mode, and the rupture distance of 1.0 mm was used. The peak force was measured in grams. The beads were tested each time and 3 replications were applied for each treatment[11, 20].

Point 6: Line 118-122 is not clear. Provide details on how bacterial counts were conducted. A 0, 15, 30, 60 and 90 days At 4 and 25°C shelflife study was conducted but justification was given in the introduction. Please include some justification in the introduction.  

Response 6: The bacteria counts were performed after storage for periods of 0, 15, 30, 60, and 90 days at 4 °C and 25 °C in yogurt samples in both encapsulated bacteria. To determine the viable counts of the bacteria, test sample contents were centrifuged (3000 g for 10 min at 4 °C) and the yogurt were dissolved by re-suspending in 9.0 mL of phosphate buffer (0.1 M, pH 7.0) followed by gentle shaking at room temperature for 15 min. The number of released cells by dilution was determined by spread plate count using MRS-salicin agar as described previously [27-28].

Point 7: Line 171: Is this the moisture content of the microcapsules or the bacteria.

Response 7: Yes the moisture content of the capsules. 

Point 8: Line 190: How was correlation determined, please include this methods under the statistical analysis on line 123.

Response 8: Statistical analysis was performed using the Pearson product moment correlation coefficient and SPSS 13.0 software (SPSS, Inc., Chicago, IL, USA)…

Point 9: Comments: Please reorder the presentation of the materials and methods based on the following suggestions: Subheading 2.1, 2.5, 2.3, 2.4, 2.6, 2.2

Response 9: It is reviewed and highlighted in the text.

Point 10: Line 297. How was the stability of the bacterial determined?

Response 10: It’s the viability. It’s corrected in the text and highlighted.

The viability of bacteria under low…

Reviewer 2 Report

The manuscript deals with the microencapsulation of probiotic bacteria with arrowroot starch for its viability in gastrointestinal tract and in yoghurt.

 Title

The title should reflect the use of arrowroot starch, maltodextrin and whey protein.

 Materials and methods

Line 104-“2.5.- Microencapsulation and coating procedures”, this section should be presented first and then the performed analyses.

Line 102- Please replace “10.0 kv” by “10.0 kV”.

Table 1- why these mass combinations??Please justify.

 Results and discussion

Line 126- “3. Results and discussion” and then line 247- “4. Discussion”??? Please revise in accordance.

Line 130- “The  strains  W.  paramesenteroides was the most constant in count at pH 2.0 (Log 7.89 cfu/mL) and 3.0 (Log 8.97 cfu/mL) respectively, whereas in free-cells, it was about Log 7.74 cfu/mL at pH 2 and Log 8.83 cfu/mL at pH 3.”???Table 2 presents other values. Please check your values and revise in accordance.

Table 3- Confusing, please align the text in 2nd column for each different experiment.

Figures 1 and 2- Please add figures with the same size and also align them correctly. Moreover, figures should be identified correctly. The pictures also present several artifacts and should be improved.

Figure 1 caption- Please format scientific names in italic.

Line 224- “from 0.5 to 1 mm. Thus, sieves of 1 mm, 500 mm and 150 mm were used to separate the different sizes.”???500 mm???150 mm??

Line 286- “The drying process may provoke these structures, which meant that fast…”???Please rephrase.

 References

Please format scientific names in italic.

Author Response

Open Review

Comments and Suggestions for Authors

The manuscript deals with the microencapsulation of probiotic bacteria with arrowroot starch for its viability in gastrointestinal tract and in yoghurt.

 Title

 "Viability of 4 Probiotic Bacteria Microencapsulated with Arrowroot Starch in the simulated Gastrointestinal Tract (GIT) and Yoghurt".

Materials and methods

Point 1: Line 104-“2.5.- Microencapsulation and coating procedures”, this section should be presented first and then the performed analyses.

Response 1: The extrusion technique of microencapsulation was derived from Kailasapathy et al. (2002) by using arrowroot as the supporting matrix. To form beads, the arrowroot solution was extruded into a previous sterile whey solution and stirred. The beads were sieved off from the solution and washed with sterile distilled water.

Wall materials were composed of arrowroot starch, maltodextrin, and whey protein for the microencapsulation of the bacteria. Chemical reagents of analytical grade were used for various characteristic tests of encapsulated bacteria. An aqueous phase was prepared by dissolving whey protein and maltodextrin or arrowroot starch (maltodextrin or arrowroot starch mix with whey protein as described in table 1) in deionized water at ice batch using a homogenizer at 3000 rpm for 5 minutes. When total solution was obtained, bacteria were added gently to the solution. The mixture was emulsified using a homogenizer at 10000 rpm for 6 minutes and 500 psi (MFG Company, Chicago, IL)[10, 21-22]. Finally, the samples were freeze-dried in a lyophilizer (Freeze Dryer Lyobeta 25) under vacuum at -40 °C for 20 hours. Freeze-dried bacteria samples were immediately transferred to sample bottles, which were sealed and stored in at -20 °C [23-24].

Point 2: Line 102- Please replace “10.0 kv” by “10.0 kV”.

Response 2: Scanning Electron Microscopy (SEM) at 10.0 kV (Hitachi S-3400, Germany).

Point 3: Table 1- why these mass combinations??Please justify.

Response 3 : This is the composition of the coating in detail.

Results and discussion

Point 4: Line 126- “3. Results and discussion” and then line 247- “4. Discussion”??? Please revise in accordance.

Response 4: It’s reviewed and highlighted in the text.

Point 5: Line 130- “The  strains  W.  paramesenteroides was the most constant in count at pH 2.0 (Log 7.89 cfu/mL) and 3.0 (Log 8.97 cfu/mL) respectively, whereas in free-cells, it was about Log 7.74 cfu/mL at pH 2 and Log 8.83 cfu/mL at pH 3.”???Table 2 presents other values. Please check your values and revise in accordance.

Response 5: The strains W. paramesenteroides was the most constant in count at pH 2.0 (log 7.89 cfu/ml) and pH 3.0 (log 7.74 cfu/ml) respectively, whereas in free-cells, it was about log 8.97 cfu/ml at pH 2 and log 8.83 cfu/ml at pH 3.

Point 6: Table 3- Confusing, please align the text in 2nd column for each different experiment.

Response 6: It’s reviewed and highlighted in the text.

Point 7: Figures 1 and 2- Please add figures with the same size and also align them correctly. Moreover, figures should be identified correctly. The pictures also present several artifacts and should be improved.

Response 7: It’s reviewed and highlighted in the text.

Point 8: Figure 1 caption- Please format scientific names in italic.

Response 8: It’s reviewed and highlighted in the text.

Point 9: Line 224- “from 0.5 to 1 mm. Thus, sieves of 1 mm, 500 mm and 150 mm were used to separate the different sizes.”???500 mm???150 mm??

Response 9: from 0.5 to 1 mm. Thus, sieves of 1 mm were used to separate the different sizes.

Point 10: Line 286- “The drying process may provoke these structures, which meant that fast…”???Please rephrase.

Response 10: These structures may be established by the fast drying process, which was led to crustiness and the diffusion of the water was reduced on top of the beads; in addition, a high pressure established surface [43-44].

Point 11: References

Please format scientific names in italic.

Response 11: It’s reviewed as requested and highlighted in the text.

Reviewer 3 Report

Manuscript foods-509899 presents an important procedure for the microencapsulation of probiotic bacteria with arroroot starch for its viability in gastrointensinal tract and in a milk based product such as yoghurt. I have some suggestions for authors to improve their work. These follow the text sequence:

-Graphical abstract

A graphical abstract presenting the main findings in short words would be of importance.

-Abstract

Lines 16-18. When authors mention ''significance or insignificance'' the p value should follow afterwards.

-Introduction

Lines 35-38. The relative work ''Coatings 2018, 8(12), 423; https://doi.org/10.3390/coatings8120423'', presenting a bio-functional product should be incorporated.

Lines 60-62.''..were colonized.....reduced the redox status....increased the total...''.

-Materials and Methods

Line 79 and elsewhere. Change ''they'' to ''these''.

Line 85. A reference should be added after the word''literature''.

Line 90.''Sampling was carried out....''.

Lines 111, 113 and elsewhere. Add space between degrees of Celsious.

Lines 123125. The authors should mention Pearson's correlation since they used it in the study.

-Results and Discussion

Lines 135-143 and elsewhere. ''log cfu/ml''.

-Table 3

A footnote about superscripts definition is missing.

Line 222.''...were performed using optical microscope''.

-Discussion

Line 255.''..of short chain fatty acids...''.

Based on the above, I suggest a minor revision of the present work.

Author Response

Open Review

Comments and Suggestions for Authors

Manuscript foods-509899 presents an important procedure for the microencapsulation of probiotic bacteria with arrowroot starch for its viability in gastrointensinal tract and in a milk based product such as yoghurt. I have some suggestions for authors to improve their work. These follow the text sequence:

-Abstract

Point 1: Lines 16-18. When authors mention ''significance or insignificance'' the p value should follow afterwards.

Response 1: It’s reviewed and highlighted in the text too.

-Introduction

Point 2: Lines 35-38. The relative work ''Coatings 2018, 8(12), 423; https://doi.org/10.3390/coatings8120423'', presenting a bio-functional product should be incorporated.

Response 2: It’s reviewed and done as requested.

Point 3: Lines 60-62.''..were colonized.....reduced the redox status....increased the total...''.

Response 3: It’s reviewed and done as requested.

-Materials and Methods

Point 4: Line 79 and elsewhere. Change ''they'' to ''these''.

Response 4: It’s reviewed and done as requested.

Point 5: Line 85. A reference should be added after the word''literature''.

Response 5: It’s reviewed and done as requested (Guam et al., )

Point 6: Line 90.''Sampling was carried out....''.

Response 6: It’s reviewed and done as requested.

Point 7: Lines 111, 113 and elsewhere. Add space between degrees of Celsious.

Response 7: It’s reviewed through all the text as requested and highlighted too.

Point 8: Lines 123125. The authors should mention Pearson's correlation since they used it in the study.

Response 8: It’s reviewed and done as requested.

-Results and Discussion

Point 9: Lines 135-143 and elsewhere. ''log cfu/ml''.

Response 9: It’s reviewed and done as requested.

Point 10: Table 3: A footnote about superscripts definition is missing.

Response 10: It is added in the text and highlighted.

Point 11: Line 222.''...were performed using optical microscope''.

Response 11: It is added in the text and highlighted.

-Discussion

Point 12: Line 255.''..of short chain fatty acids...''.

Response 12: It is added in the text and highlighted.

Based on the above, I suggest a minor revision of the present work.

Thanks so much Dear reviewer for your appreciation of our research, we had followed all your suggestions and corrections plus other’s work too.

Round  2

Reviewer 2 Report

SEM pictures should be resized to present the same dimensions.

References

Please format scientific names in italic.

Author Response

Comments and Suggestions for Authors

Point 1: SEM pictures should be resized to present the same dimensions.

Response 1: It is reviewed and highlighted in the text. The new size of all the pictures is 2.2”x2.92”, even though they look different from each other but it’s the same size. Thanks for your suggestion and appreciation. We look forward to hear from you.

References

Point 2: Please format scientific names in italic.

Response 2: It is reviewed and highlighted in the text. Example:

Ilha, E. C.; Da Silva, T.; Lorenz, J. G.; de Oliveira Rocha, G.; Sant’Anna, E. S., Lactobacillus paracasei isolated 454 from grape sourdough: acid, bile, salt, and heat tolerance after spray drying with skim milk and cheese 455 whey. Eur. Food Res. Technol. 2015, 240 (5), 977-984. https://doi.org/10.1111/j.1472-765X.2005.01778.x.